# Spike-Train Level Backpropagation for Training Deep Recurrent Spiking Neural Networks

**Wenrui Zhang**
University of California, Santa Barbara
Santa Barbara, CA 93106
wenruizhang@ucsb.edu

**Peng Li**
University of California, Santa Barbara
Santa Barbara, CA 93106
lip@ucsb.edu

## Abstract

Spiking neural networks (SNNs) well support spatio-temporal learning and energy-efficient event-driven hardware neuromorphic processors. As an important class of SNNs, recurrent spiking neural networks (RSNNs) possess great computational power. However, the practical application of RSNNs is severely limited by challenges in training. Biologically-inspired unsupervised learning has limited capability in boosting the performance of RSNNs. On the other hand, existing backpropagation (BP) methods suffer from high complexity of unfolding in time, vanishing and exploding gradients, and approximate differentiation of discontinuous spiking activities when applied to RSNNs. To enable supervised training of RSNNs under a well-defined loss function, we present a novel Spike-Train level RSNNs Backpropagation (ST-RSBP) algorithm for training deep RSNNs. The proposed ST-RSBP directly computes the gradient of a rate-coded loss function defined at the output layer of the network w.r.t tunable parameters. The scalability of ST-RSBP is achieved by the proposed spike-train level computation during which temporal effects of the SNN is captured in both the forward and backward pass of BP. Our ST-RSBP algorithm can be broadly applied to RSNNs with a single recurrent layer or deep RSNNs with multiple feedforward and recurrent layers. Based upon challenging speech and image datasets including TI46 [25], N-TIDIGITS [3], Fashion-MNIST [40] and MNIST, ST-RSBP is able to train SNNs with an accuracy surpassing that of the current state-of-the-art SNN BP algorithms and conventional non-spiking deep learning models.

## 1 Introduction

In recent years, deep neural networks (DNNs) have demonstrated outstanding performance in natural language processing, speech recognition, visual object recognition, object detection, and many other domains [6, 14, 21, 36, 13]. On the other hand, it is believed that biological brains operate rather differently [17]. Neurons in artificial neural networks (ANNs) are characterized by a single, static, and continuous-valued activation function. More biologically plausible spiking neural networks (SNNs) compute based upon discrete spike events and spatio-temporal patterns while enjoying rich coding mechanisms including rate and temporal codes [11]. There is theoretical evidence supporting that SNNs possess greater computational power over traditional ANNs [11]. Moreover, the event-driven nature of SNNs enables ultra-low-power hardware neuromorphic computing devices [7, 2, 10, 28].

Backpropagation (BP) is the workhorse for training deep ANNs [22]. Its success in the ANN world has made BP a target of intensive research for SNNs. Nevertheless, applying BP to biologically more plausible SNNs is nontrivial due to the necessity in dealing with complex neural dynamics and non-differentiability of discrete spike events. It is possible to train an ANN and then convert it to an SNN [9, 10, 16]. However, this suffers from conversion approximations and gives up the

opportunity in exploring SNNs' temporal learning capability. One of the earliest attempts to bridge the gap between discontinuity of SNNs and BP is the SpikeProp algorithm [5]. However, SpikeProp is restricted to single-spike learning and has not yet been successful in solving real-world tasks.

Recently, training SNNs using BP under a firing rate (or activity level) coded loss function has been shown to deliver competitive performances [23, 39, 4, 33]. Nevertheless, [23] does not consider the temporal correlations of neural activities and deals with spiking discontinuities by treating them as noise. [33] gets around the non-differentiability of spike events by approximating the spiking process via a probability density function of spike state change. [39], [4], and [15] capture the temporal effects by performing backpropagation through time (BPTT) [37]. Among these, [15] adopts a smoothed spiking threshold and a continuous differentiable synaptic model for gradient computation, which is not applicable to widely used spiking neuron models such as the leaky integrate-and-fire (LIF) model. Similar to [23], [39] and [4] compute the error gradient based on the continuous membrane waveforms resulted from smoothing out all spikes. In these approaches, computing the error gradient by smoothing the microscopic membrane waveforms may lose the sight of the all-or-none firing characteristics of the SNN that defines the higher-level loss function and lead to inconsistency between the computed gradient and target loss, potentially degrading training performance [19].

Most existing SNN training algorithms including the aforementioned BP works focus on feedforward networks. Recurrent spiking neural networks (RSNNs), which are an important class of SNNs and are especially competent for processing temporal signals such as time series or speech data [12], deserve equal attention. The liquid State Machine (LSM) [27] is a special RSNN which has a single recurrent reservoir layer followed by one readout layer. To mitigate training challenges, the reservoir weights are either fixed or trained by unsupervised learning like spike-timing-dependent plasticity (STDP) [29] with only the readout layer trained by supervision [31, 41, 18]. The inability in training the entire network with supervision and its architectural constraints, e.g. only admitting one reservoir and one readout, limit the performance of LSM. [4] proposes an architecture called long short-term memory SNNs (LSNNs) and trains it using BPTT with the aforementioned issue on approximate gradient computation. When dealing with training of general RSNNs, in addition to the difficulties encountered in feedforward SNNs, one has to cope with added challenges incurred by recurrent connections and potential vanishing/exploding gradients.

This work is motivated by: **1)** lack of powerful supervised training of general RSNNs, and **2)** an immediate outcome of 1), i.e. the existing SNN research has limited scope in exploring sophisticated learning architectures like deep RSNNs with multiple feedforward and recurrent layers hybridized together. As a first step towards addressing these challenges, we propose a novel biologically non-plausible Spike-Train level RSNNs Backpropagation (ST-RSBP) algorithm which is applicable to RSNNs with an arbitrary network topology and achieves the state-of-the-art performances on several widely used datasets. The proposed ST-RSBP employs spike-train level computation similar to what is adopted in the recent hybrid macro/micro level BP (HM2-BP) method for feedforward SNNs [19], which demonstrates encouraging performances and outperforms BPTT such as the one implemented in [39].

ST-RSBP is rigorously derived and can handle arbitrary recurrent connections in various RSNNs. While capturing the temporal behavior of the RSNN at the spike-train level, ST-RSBP directly computes the gradient of a rate-coded loss function w.r.t tunable parameters without incurring approximations resulted from altering and smoothing the underlying spiking behaviors. ST-RSBP is able to train RSNNs without costly unfolding the network through time and performing BP time point by time point, offering faster training and avoiding vanishing/exploding gradients for general RSNNs. Moreover, as mentioned in Section 2.2.1 and 2.3 of the Supplementary Materials, since ST-RSBP more precisely computes error gradients than HM2-BP [19], it can achieve better results than HM2-BP even on the feedforward SNNs.

We apply ST-RSBP to train several deep RSNNs with multiple feedforward and recurrent layers to demonstrate the best performances on several widely adopted datasets. Based upon challenging speech and image datasets including TI46 [25], N-TIDIGITS [3] and Fashion-MNIST [40], ST-RSBP trains RSNNs with an accuracy noticeably surpassing that of the current state-of-the-art SNN BP algorithms and conventional non-spiking deep learning models and algorithms. Furthermore, ST-RSBP is also evaluated on feedforward spiking convolutional neural networks (spiking CNNs) with the MNIST dataset and achieves 99.62% accuracy, which is the best among all SNN BP rules.

# 2 Background

## 2.1 SNN Architectures and Training Challenges

Fig. 1A shows two SNN architectures often explored in neuroscience: single layer (top) and liquid state machine (bottom) networks for which different mechanisms have been adopted for training. However, typically spike timing dependent plasticity (STDP) [29] and winner-take-all (WTA) [8] are only for unsupervised training and have limited performance. WTA and other supervised learning rules [31, 41, 18] can only be applied to the output layer, obstructing adoption of more sophisticated deep architectures.

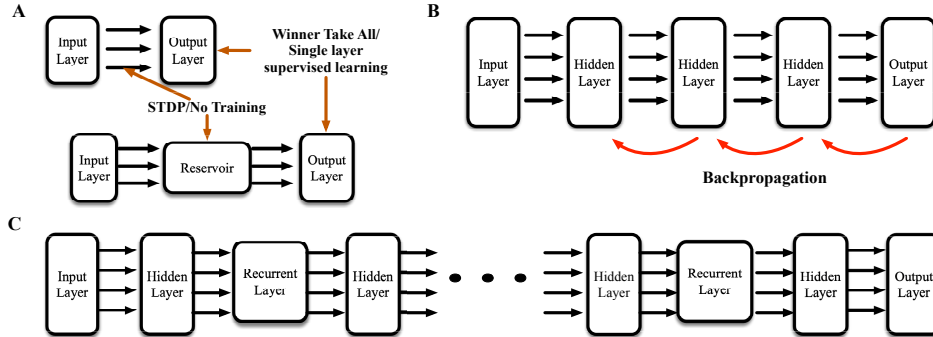

Figure 1: Various SNN networks: (A) one layer SNNs and liquid state machine; (B) multi-layer feedforward SNNs; (C) deep hybrid feedforward/recurrent SNNs.

While bio-inspired learning mechanisms are yet to demonstrate competitive performance for challenging real-life tasks, there has been much recent effort aiming at improving SNN performance with supervised BP. Most existing SNN BP methods are only applicable to multi-layer feedforward networks as shown in Fig. 1B. Several such methods have demonstrated promising results [23, 39, 19, 33]. Nevertheless, these methods are not applicable to complex deep RSNNs such as the hybrid feedforward/recurrent networks shown in Fig. 1C, which are the target of this work. Backpropagation through time (BPTT) in principle may be applied to training RSNNs [4], but bottlenecked with several challenges in: (1) unfolding the recurrent connections through time, (2) back propagating errors over both time and space, and (3) back propagating errors over non-differentiable spike events.

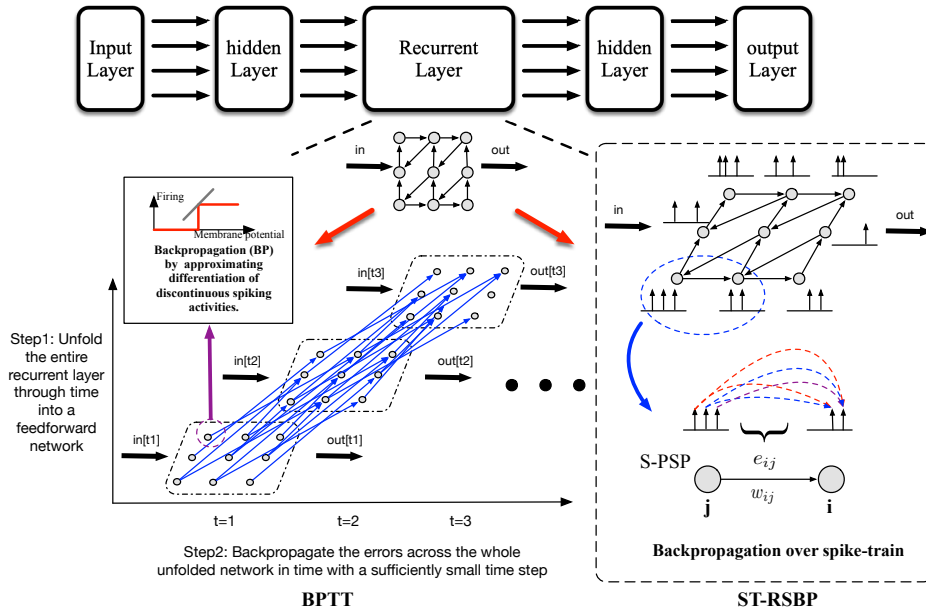

Figure 2: Backpropagation in recurrent SNNs: BPTT vs. ST-RSBP.

Fig. 2 compares BPTT and ST-RSBP, where we focus on a recurrent layer since feedforward layer can be viewed as a simplified recurrent layer. To apply BPTT, one shall first unfold a RSNN in time

to convert it into a larger feedforward network without recurrent connections. The total number of layers in the feedforward network is increased by a factor equal to the number of times the RSNN is unfolded, and hence can be very large. Then, this unfolded network is integrated in time with a sufficiently small time step to capture dynamics of the spiking behavior. BP is then performed spatio-temproally layer-by-layer across the unfolded network based on the same time stepsize used for integration as shown in Fig. 2. In contrast, the proposed ST-RSBP does not convert the RSNN into a larger feedforward SNN. The forward pass of BP is based on time-domain integration of the RSNN of the original size. Following the forward pass, importantly, the backward pass of BP is not conducted point by point in time, but instead, much more efficiently on the spike-train level. We make use of Spike-train Level Post-synaptic Potentials (S-PSPs) discussed in Section 2.2 to capture temporal interactions between any pair of pre/post-synaptic neurons. ST-RSBP is more scalable and has the added benefits of avoiding exploding/vanishing gradients for general RSNNs.

## 2.2 Spike-train Level Post-synaptic Potential (S-PSP)

S-PSP captures the spike-train level interactions between a pair of pre/post-synaptic neurons. Note that each neuron fires whenever its post-synaptic potential reaches the firing threshold. The accumulated contributions of the pre-synaptic neuron $j$'s spike train to the (normalized) post-synaptic potential of the neuron $i$ right before all the neuron $i$'s firing times is defined as the (normalized) S-PSP from the neuron $j$ to the neuron $i$ as in (6) in the Supplementary Materials. The S-PSP $e_{ij}$ characterizes the aggregated effect of the spike train of the neuron $j$ on the membrane potential of the neuron $i$ and its firing activities. S-PSPs allow consideration of the temporal dynamics and recurrent connections of an RSNN across all firing events at the spike-train level without expensive unfolding through time and backpropagation time point by time point.

The sum of the weighted S-PSPs from all pre-synaptic neurons of the neuron $i$ is defined as the **total post-synaptic potential** (**T-PSP**) $a_i$. $a_i$ is the post-synaptic membrane potential accumulated right before all firing times and relates to the firing count $o_i$ via the firing threshold $\nu$ [19]:

$$a_i = \sum_j w_{ij}\, e_{ij}, \qquad o_i = g(a_i) \approx \frac{a_i}{\nu}. \tag{1}$$

$a_i$ and $o_i$ are analogous to the pre-activation and activation in the traditional ANNs, respectively, and $g(\cdot)$ can be considered as an activation function converting the T-PSP to the output firing count.

A detailed description of S-PSP and T-PSP can be found in Section 1 in the Supplementary Materials.

## 3 Proposed Spike-Train level Recurrent SNNs Backpropagation (ST-RSBP)

We use the generic recurrent spiking neural network with a combination of feedforward and recurrent layers of Fig. 2 to derive ST-RSBP. For the spike-train level activation of each neuron $l$ in the layer $k+1$, (1) is modified to include the recurrent connections explicitly if necessary:

$$a_l^{k+1} = \sum_{j=1}^{N_k} w_{lj}^{k+1}\, e_{lj}^{k+1} + \sum_{p=1}^{N_{k+1}} w_{lp}^{k+1}\, e_{lp}^{k+1}, \qquad o_l^{k+1} = g(a_l^{k+1}) \approx \frac{a_l^{k+1}}{\nu^{k+1}}. \tag{2}$$

$N_{k+1}$ and $N_k$ are the number of neurons in the layers $k+1$ and $k$, $w_{lj}^{k+1}$ is the feedforward weight from the neuron $j$ in the layer $k$ to the neuron $l$ in the layer $k+1$, $w_{lp}^{k+1}$ is the recurrent weight from the neuron $p$ to the neuron $l$ in the layer $k+1$, which is non-existent if the layer $k+1$ is feedforward, $e_{lj}^{k+1}$ and $e_{lp}^{k+1}$ are the corresponding S-PSPs, $\nu^{k+1}$ is the firing threshold at the layer $k+1$, $o_l^{k+1}$ and $a_l^{k+1}$ are the firing count and pre-activation (T-PSP) of the neuron $l$ at the layer $k+1$, respectively.

The rate-coded loss is defined at the output layer as:

$$E = \frac{1}{2}\|\boldsymbol{o} - \boldsymbol{y}\|_2^2 = \frac{1}{2}\|\frac{\boldsymbol{a}}{\nu} - \boldsymbol{y}\|_2^2, \tag{3}$$

where $\boldsymbol{y}$, $\boldsymbol{o}$ and $\boldsymbol{a}$ are vectors of the desired output neuron firing counts (labels) and actual firing counts, and the T-PSPs of the output neurons, respectively. Differentiating (3) with respect to each trainable weight $w_{ij}^k$ incident upon the layer $k$ leads to:

$$\frac{\partial E}{\partial w_{ij}^k} = \frac{\partial E}{\partial a_i^k}\frac{\partial a_i^k}{\partial w_{ij}^k} = \delta_i^k \frac{\partial a_i^k}{\partial w_{ij}^k} \qquad \text{with} \quad \delta_i^k = \frac{\partial E}{\partial a_i^k}, \tag{4}$$

where $\delta_i^k$ and $\frac{\partial a_i^k}{\partial w_{ij}^k}$ are referred to as the ***back propagated error*** and ***differentiation of activation***, respectively, for the neuron $i$. ST-RSBP updates $w_{ij}^k$ by $\Delta w_{ij}^k = \eta \frac{\partial E}{\partial w_{ij}^k}$, where $\eta$ is a learning rate.

We outline the key component of derivation of ST-RSBP: the back propagated errors. The full derivation of ST-RSBP is presented in Section 2 of the Supplementary Materials.

### 3.1 Outline of the Derivation of Back Propagated Errors

#### 3.1.1 Output Layer

If the layer $k$ is the output, the back propagated error of the neuron $i$ is given by differentiating (3):

$$\delta_i^k = \frac{\partial E}{\partial a_i^k} = \frac{(o_i^k - y_i^k)}{\nu^k}, \tag{5}$$

where $o_i^k$ is the actual firing count, $y_i^k$ the desired firing count (label), and $a_i^k$ the T-PSP.

#### 3.1.2 Hidden Layers

At each hidden layer $k$, the chain rule is applied to determine the error $\delta_i$ for the neuron $i$:

$$\delta_i^k = \frac{\partial E}{\partial a_i^k} = \sum_{l=1}^{N_{k+1}} \frac{\partial E}{\partial a_l^{k+1}} \frac{\partial a_l^{k+1}}{\partial a_i^k} = \sum_{l=1}^{N_{k+1}} \delta_l^{k+1} \frac{\partial a_l^{k+1}}{\partial a_i^k}. \tag{6}$$

Define two error vectors $\boldsymbol{\delta}^{k+1}$ and $\boldsymbol{\delta}^k$ for the layers $k+1$ and $k$ : $\boldsymbol{\delta}^{k+1} = [\delta_1^{k+1}, \cdots, \delta_{N_{k+1}}^{k+1}]$, and $\boldsymbol{\delta}^k = [\delta_1^k, \cdots, \delta_{N_k}^k]$, respectively. Assuming $\boldsymbol{\delta}^{k+1}$ is given, which is the case for the output layer, the goal is to back propagate from $\boldsymbol{\delta}^{k+1}$ to $\boldsymbol{\delta}^k$. This entails to compute $\frac{\partial a_l^{k+1}}{\partial a_i^k}$ in (6).

**[Backpropagation from a Hidden Recurrent Layer]** Now consider the case that the errors are back propagated from a recurrent layer $k+1$ to its preceding layer $k$. Note that the S-PSP $e_{lj}$ from any pre-synaptic neuron $j$ to a post-synaptic neuron $l$ is a function of both the rate and temporal information of the pre/post-synaptic spike trains, which can be made explicitly via some function $f$:

$$e_{lj} = f(o_j, o_l, \boldsymbol{t}_j^{(f)}, \boldsymbol{t}_l^{(f)}), \tag{7}$$

where $o_j$, $o_l$, $\boldsymbol{t}_j^{(f)}$, $\boldsymbol{t}_l^{(f)}$ are the pre/post-synaptic firing counts and firing times, respectively.

Now based on (2), $\frac{\partial a_l^{k+1}}{\partial a_i^k}$ is split also into two summations:

$$\frac{\partial a_l^{k+1}}{\partial a_i^k} = \sum_{j}^{N_k} w_{lj}^{k+1} \frac{de_{lj}^{k+1}}{da_i^k} + \sum_{p}^{N_{k+1}} w_{lp}^{k+1} \frac{de_{lp}^{k+1}}{da_i^k}, \tag{8}$$

where the first summation sums over all pre-synaptic neurons in the previous layer $k$ while the second sums over the pre-synaptic neurons in the current recurrent layer as illustrated in Fig. 3.

On the right side of (8), $\frac{de_{lj}^{k+1}}{da_i^k}$ is given by:

$$\frac{de_{lj}^{k+1}}{da_i^k} = \begin{cases} \frac{1}{\nu^k} \frac{\partial e_{li}^{k+1}}{\partial o_i^k} + \frac{1}{\nu^{k+1}} \frac{\partial e_{lj}^{k+1}}{\partial o_l^{k+1}} \frac{\partial a_l^{k+1}}{\partial a_i^k} & j = i \\ \frac{1}{\nu^{k+1}} \frac{\partial e_{lj}^{k+1}}{\partial o_l^{k+1}} \frac{\partial a_l^{k+1}}{\partial a_i^k} & j \neq i. \end{cases} \tag{9}$$

$\nu^k$ and $\nu^{k+1}$ are the firing threshold voltages for the layers $k$ and $k+1$, respectively, and we have used that $o_i^k \approx a_i^k / \nu^k$ and $o_l^{k+1} \approx a_l^{k+1} / \nu^{k+1}$ from (1). Importantly, the last term on the right side of (9) exists due to $e_{lj}^{k+1}$'s dependency on the post-synaptic firing rate $o_l^{k+1}$ per (7) and

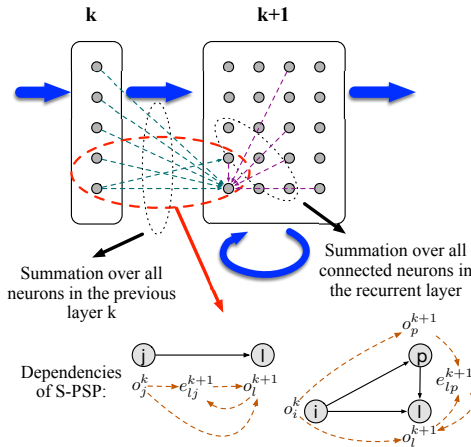

Figure 3: Connections for a recurrent layer neuron and the dependencies among its S-PSPs.

$o_l^{k+1}$'s further dependency on the pre-synaptic activation $o_i^k$ (hence pre-activation $a_i^k$), as shown in Fig. 3.

On the right side of (8), $\frac{de_{lp}^{k+1}}{da_i^k}$ is due to the recurrent connections within the layer $k+1$:

$$\frac{de_{lp}^{k+1}}{da_i^k} = \frac{1}{\nu^{k+1}} \frac{\partial e_{lp}^{k+1}}{\partial o_l^{k+1}} \frac{\partial a_l^{k+1}}{\partial a_i^k} + \frac{1}{\nu^{k+1}} \frac{\partial e_{lp}^{k+1}}{\partial o_p^{k+1}} \frac{\partial a_p^{k+1}}{\partial a_i^k}. \tag{10}$$

The first term on the right side of (10) is due to $e_{lp}^{k+1}$'s dependency on the post-synaptic firing rate $o_l^{k+1}$ per (7) and $o_l^{k+1}$'s further dependence on the pre-synaptic activation $o_i^k$ (hence pre-activation $a_i^k$). Per (7), it is important to note that the second term exists because $e_{lp}^{k+1}$'s dependency on the pre-synaptic firing rate $o_p^{k+1}$, which further depends on $o_i^k$ (hence pre-activation $a_i^k$), as shown in Fig. 3.

Putting (8), (9), and (10) together leads to:

$$\left(1 - \frac{1}{\nu^{k+1}} \left( \sum_j^{N_k} w_{lj}^{k+1} \frac{\partial e_{lj}^{k+1}}{\partial o_l^{k+1}} + \sum_p^{N_{k+1}} w_{lp}^{k+1} \frac{\partial e_{lp}^{k+1}}{\partial o_l^{k+1}} \right) \right) \frac{\partial a_l^{k+1}}{\partial a_i^k}$$
$$= w_{li}^{k+1} \frac{1}{\nu^k} \frac{\partial e_{li}^{k+1}}{\partial o_i^k} + \sum_p^{N_{k+1}} w_{lp}^{k+1} \frac{1}{\nu^{k+1}} \frac{\partial e_{lp}^{k+1}}{\partial o_p^{k+1}} \frac{\partial a_p^{k+1}}{\partial a_i^k}. \tag{11}$$

It is evident that all $N_{k+1} \times N_k$ partial derivatives involving the recurrent layer $k+1$ and its preceding layer $k$, i.e. $\frac{\partial a_l^{k+1}}{\partial a_i^k}, l = [1, N_{k+1}], i = [1, N_k]$, form a coupled linear system via (11), which is written in a matrix form as:

$$\boldsymbol{\Omega}^{k+1,k} \cdot \boldsymbol{P}^{k+1,k} = \boldsymbol{\Phi}^{k+1,k} + \boldsymbol{\Theta}^{k+1,k} \cdot \boldsymbol{P}^{k+1,k}, \tag{12}$$

where $\boldsymbol{P}^{k+1,k} \in \mathbb{R}^{N_{k+1} \times N_k}$ contains all the desired partial derivatives, $\boldsymbol{\Omega}^{k+1,k} \in \mathbb{R}^{N_{k+1} \times N_{k+1}}$ is diagonal, $\boldsymbol{\Theta}^{k+1,k} \in \mathbb{R}^{N_{k+1} \times N_{k+1}}$, $\boldsymbol{\Phi}^{k+1,k} \in \mathbb{R}^{N_{k+1} \times N_k}$, and the detailed definitions of all these matrices can be found in Section 2.1 of the Supplementary Materials.

Solving the linear system in (12) gives all $\frac{\partial a_i^{k+1}}{\partial a_j^k}$:

$$\boldsymbol{P}^{k+1,k} = (\boldsymbol{\Omega}^{k+1,k} - \boldsymbol{\Theta}^{k+1,k})^{-1} \cdot \boldsymbol{\Phi}^{k+1,k}. \tag{13}$$

Note that since $\boldsymbol{\Omega}$ is a diagonal matrix, the cost in factoring the above linear system can be reduced by approximating the matrix inversion using a first-order Taylor's expansion without matrix factorization. Error propagation from the layer $k+1$ to layer $k$ of (6) is cast in the matrix form: $\boldsymbol{\delta}^k = \boldsymbol{P}^T \cdot \boldsymbol{\delta}^{k+1}$.

**[Backpropagation from a Hidden Feedforward Layer]** The much simpler case of backpropagating errors from a feedforward layer $k+1$ to its preceding layer $k$ is described in Section 2.1 of the Supplementary Materials.

The complete ST-RSBP algorithm is summarized in Section 2.4 in the Supplementary Materials.

# 4 Experiments and Results

## 4.1 Experimental Settings

All reported experiments below are conducted on an NVIDIA Titan XP GPU. The experimented SNNs are based on the LIF model and weights are randomly initialized by following the uniform distribution $U[-1, 1]$. Fixed firing thresholds are used in the range of $5mV$ to $20mV$ depending on the layer. Exponential weight regularization [23], lateral inhibition in the output layer [23] and Adam [20] as the optimizer are adopted. The parameters like the desired output firing counts, thresholds and learning rates are empirically tuned. Table 1 lists the typical constant values adopted in the proposed ST-RSBP learning rule in our experiments. The simulation step size is set to 1 ms. The batch size is 1 which means ST-RSBP is applied after each training sample to update the weights.

Using three speech datasets and two image dataset, we compare the proposed ST-RSBP with several other methods which either have the best previously reported results on the same datasets or represent the current state-of-the-art performances for training SNNs. Among these, HM2-BP [19] is the best reported BP algorithm for feedforward SNNs based on LIF neurons. ST-RSBP is evaluated

Table 1: Parameters settings

| Parameter | Value | Parameter | Value |
|---|---|---|---|
| Time Constant of Membrane Voltage $\tau_m$ | 64 ms | Threshold $\nu$ | 10 mV |
| Time Constant of Synapse $\tau_s$ | 8 ms | Synaptic Time Delay | 1 ms |
| Refractory Period | 2 ms | Reset Membrane Voltage $V_{reset}$ | 0 mV |
| Desired Firing Count for Target Neuron | 35 | Learning Rate $\eta$ | 0.001 |
| Desired Firing Count for Non-Target Neuron | 5 | Batch Size | 1 |

using RSNNs of multiple feedforward and recurrent layers with full connections between adjacent layers and sparse connections inside the recurrent layers. The network models of all other BP methods we compare with are fully connected feedforward networks. The liquid state machine (LSM) networks demonstrated below have sparse input connections, sparse reservoir connections, and a fully connected readout layer. Since HM2-BP cannot train recurrent networks, we compare ST-RSBP with HM2-BP using models of a similar number of tunable weights. Moreover, we also demonstrate that ST-RSBP achieves the best performance among several state-of-the-art SNN BP rules evaluated on the same or similar spiking CNNs. Each experiment reported below is repeated five times to obtain the mean and standard deviation (stddev) of the accuracy.

## 4.2 TI46-Alpha Speech Dataset

TI46-Alpha is the full alphabets subset of the TI46 Speech corpus [25] and contains spoken English alphabets from 16 speakers. There are 4,142 and 6,628 spoken English examples in 26 classes for training and testing, respectively. The continuous temporal speech waveforms are first preprocessed by the Lyon's ear model [26] and then encoded into 78 spike trains using the BSA algorithm [32].

Table 2: Comparison of different SNN models on TI46-Alpha

| Algorithm | Hidden Layers[a] | # Params | # Epochs | Mean | Stddev | Best |
|---|---|---|---|---|---|---|
| HM2-BP [19] | 800 | 83,200 | 138 | 89.36% | 0.30% | 89.92% |
| HM2-BP [19] | 400-400 | 201,600 | 163 | 89.83% | 0.71% | 90.60% |
| HM2-BP [19] | 800-800 | 723,200 | 174 | 90.50% | 0.45% | 90.98% |
| Non-spiking BP[b] [38] | LSM: R2000 | 52,000 | | | | 78% |
| ST-RSBP (this work) | R800 | 86,280 | 75 | 91.57% | 0.20% | 91.85% |
| ST-RSBP (this work) | 400-R400-400 | 363,313 | 57 | 93.06% | 0.21% | **93.35%** |

[a] We show the number of neurons in each feedforward/recurrent hidden layer. R represent recurrent layer.

[b] An LSM model. The state vector of the reservoir is used to train the single readout layer using BP.

Table 2 compares ST-RSBP with several other algorithms on TI46-Alpha. The result from [38] shows that only training the single readout layer of a recurrent LSM is inadequate for this challenging task, demonstrating the necessity of training all layers of a recurrent network using techniques such as ST-RSBP. ST-RSBP outperforms all other methods. In particular, ST-RSBP is able to train a three-hidden-layer RSNN with 363,313 weights to increase the accuracy from $90.98\%$ to $93.35\%$ when compared with the feedforward SNN with 723,200 weights trained by HM2-BP.

## 4.3 TI46-Digits Speech Datasest

TI46-Digits is the full digits subset of the TI46 Speech corpus [25]. It contains 1,594 training examples and 2,542 testing examples of 10 utterances for each of digits "0" to "9" spoken by 16 different speakers. The same preprocessing used for TI46-Alpha is adopted. Table 3 shows that the proposed ST-RSBP delivers a high accuracy of $99.39\%$ while outperforming all other methods including HM2-BP. On recurrent network training, ST-RSBP produces large improvements over two other methods. For instance, with 19,057 tunable weights, ST-RSBP delivers an accuracy of $98.77\%$ while [35] has an accuracy of $86.66\%$ with 32,000 tunable weights.

Table 3: Comparison of different SNN models on TI46-Digits

| Algorithm | Hidden Layers | # Params | # Epochs | Mean | Stddev | Best |
|---|---|---|---|---|---|---|
| HM2-BP [19] | 100-100 | 18,800 | 22 | | | 98.42% |
| HM2-BP [19] | 200-200 | 57,600 | 21 | | | 98.50% |
| Non-spiking BP [38] | LSM: R500 | 5,000 | | | | 78% |
| SpiLinC[a] [35] | LSM: R3200 | 32,000 | | | | 86.66% |
| ST-RSBP (this work) | R100-100 | 19,057 | 75 | 98.77% | 0.13% | 98.95% |
| ST-RSBP (this work) | R200-200 | 58,230 | 28 | 99.16% | 0.11% | 99.27% |
| ST-RSBP (this work) | 200-R200-200 | 98,230 | 23 | 99.25% | 0.13% | **99.39%** |

[a] An LSM with multiple reservoirs in parallel. Weights between input and reservoirs are trained using STDP. The excitatory neurons in the reservoir are tagged with the classes for which they spiked at a highest rate during training and are grouped accordingly. During inference, for a test pattern, the average spike count of every group of neurons tagged is examined and the tag with the highest average spike count represents the predicted class.

## 4.4 N-Tidigits Neuromorphic Speech Dataset

The N-Tidigits [3] is the neuromorphic version of the well-known speech dataset Tidigits, and consists of recorded spike responses of a 64-channel CochleaAMS1b sensor in response to audio waveforms from the original Tidigits dataset [24]. 2,475 single digit examples are used for training and the same number of examples are used for testing. There are 55 male and 56 female speakers and each of them speaks two examples for each of the 11 single digits including "oh," "zero", and the digits "1" to "9". Table 4 shows that proposed ST-RSBP achieves excellent accuracies up to 93.90%, which is significantly better than that of HM2-BP and the non-spiking GRN and LSTM in [3]. With a similar/less number of tunable weights, ST-RSBP outperforms all other methods rather significantly.

Table 4: Comparison of different models on N-Tidigits

| Algorithm | Hidden Layers | # Params | # Epoch | Mean | Stddev | Best |
|---|---|---|---|---|---|---|
| HM2-BP [19] | 250-250 | 81,250 | | | | 89.69% |
| GRN (NS[a]) [3] | 2× G200-100[b] | 109,200 | | | | 90.90% |
| Phased-LSTM (NS) [3] | 2× 250L[c] | 610,500 | | | | 91.25% |
| ST-RSBP (this work) | 250-R250 | 82,050 | 268 | 92.94% | 0.20% | 93.13% |
| ST-RSBP (this work) | 400-R400-400 | 351,241 | 287 | 93.63% | 0.27% | **93.90%** |

[a] NS represents non-spiking algorithm; [b] G represents a GRN layer; [c] L represents an LSTM layer.

Table 5: Comparison of different models on Fashion-MNIST

| Algorithm | Hidden Layers | # Params | # Epochs | Mean | Stddev | Best |
|---|---|---|---|---|---|---|
| HM2-BP [19] | 400-400 | 477,600 | 15 | | | 88.99% |
| BP [30][a] | 5× 256 | 465,408 | | | | 87.02% |
| LRA-E [30][b] | 5× 256 | 465,408 | | | | 87.69% |
| DL BP [1][a] | 3× 512 | 662,026 | | | | 89.06% |
| Keras BP[c] | 512-512 | 669706 | 50 | | | 89.01% |
| ST-RSBP (this work) | 400-R400 | 478,841 | 36 | 90.00% | 0.14% | **90.13%** |

[a] Fully connected ANN trained with the BP algorithm.
[b] Fully connected ANN with locally defined errors trained using gradient descent. Loss functions are L2 norm for hidden layers and categorical cross-entropy for the output layer.
[c] Fully connected ANN trained using the Keras package with RELU activation, categorical cross-entropy loss, and RMSProp optimizer; a dropout layer applied between each dense layer with rate of 0.2.

## 4.5 Fashion-MNIST Image Dataset

The Fashion-MNIST dataset [40] contains 28x28 grey-scale images of clothing items, meant to serve as a much more difficult drop-in replacement for the well-known MNIST dataset. It contains 60,000 training examples and 10,000 testing examples with each image falling under one of the 10 classes. Using Poisson sampling, we encode each $28 \times 28$ image into a 2D $784 \times L$ binary matrix, where $L = 400$ represents the duration of each spike sequence in $ms$, and a 1 in the matrix represents a

spike. The simulation time step is set to be $1ms$. No other preprocessing or data augmentation is applied. Table 5 shows that ST-RSBP outperforms all other SNN and non-spiking BP methods.

## 4.6 Spiking Convolution Neural Networks for the MNIST

As mentioned in Section 1, ST-RSBP can more precisely compute gradients error than HM2-BP even for the case of feedforward CNNs. We demonstrate the performance improvement of ST-RSBP over several other state-of-the-art SNN BP algorithms based on spiking CNNs using the MNIST dataset. The preprocessing steps are the same as the ones for Fashion-MNIST in Section 4.5. The spiking CNN trained by ST-RSBP consists of two $5 \times 5$ convolutional layers with a stride of 1, each followed by a $2 \times 2$ pooling layer, one fully connected hidden layer and an output layer for classification. In the pooling layer, each neuron connects to $2 \times 2$ neurons in the preceding convolutional layer with a fixed weight of $0.25$. In addition, we use elastic distortion [34] for data augmentation which is similar to [23, 39, 19]. In Table 6, we compare the results of the proposed ST-RSBP with other BP rules on similar network settings. It shows that ST-RSBP can achieve an accuracy of $99.62\%$, surpassing the best previously reported performance [19] with the same model complexity.

Table 6: Performances of Spiking CNNs on MNIST

| Algorithm | Hidden Layers | Mean | Stddev | Best |
|---|---|---|---|---|
| Spiking CNN [23] | 20C5-P2-50C5-P2-200[a] | | | 99.31% |
| STBP [39] | 15C5-P2-40C5-P2-300 | | | 99.42% |
| SLAYER [33] | 12C5-p2-64C5-p2 | 99.36% | 0.05% | 99.41% |
| HM2-BP [19] | 15C5-P2-40C5-P2-300 | 99.42% | 0.11% | 99.49% |
| ST-RSBP (this work) | 12C5-p2-64C5-p2 | 99.50% | 0.03% | 99.53% |
| ST-RSBP (this work) | 15C5-P2-40C5-P2-300 | 99.57% | 0.04% | **99.62%** |

[a] 20C5 represents convolution layer with 20 of the $5 \times 5$ filters. P2 represents pooling layer with $2 \times 2$ filters.

## 5 Discussions and Conclusion

In this paper, we present the novel spike-train level backpropagation algorithm ST-RSBP, which can transparently train all types of SNNs including RSNNs without unfolding in time. The employed S-PSP model improves the training efficiency at the spike-train level and also addresses key challenges of RSNNs training in handling of temporal effects and gradient computation of loss functions with inherent discontinuities for accurate gradient computation. The spike-train level processing for RSNNs is the starting point for ST-RSBP. After that, we have applied the standard BP principle while dealing with specific issues of derivative computation at the spike-train level.

More specifically, in ST-RSBP, the given rate-coded errors can be efficiently computed and back-propagated through layers without costly unfolding the network in time and through expensive time point by time point computation. Moreover, ST-RSBP handles the discontinuity of spikes during BP without altering and smoothing the microscopic spiking behaviors. The problem of network unfolding is dealt with accurate spike-train level BP such that the effect of all spikes are captured and propagated in an aggregated manner to achieve accurate and fast training. As such, both rate and temporal information in the SNN are well exploited during the training process.

Using the efficient GPU implementation of ST-RSBP, we demonstrate the best performances for both feedforward SNNs, RSNNs and spiking CNNs over the speech datasets TI46-Alpha, TI46-Digits, and N-Tidigits and the image dataset MNIST and Fashion-MNIST, outperforming the current state-of-the-art SNN training techniques. Moreover, ST-RSBP outperforms conventional deep learning models like LSTM, GRN, and traditional non-spiking BP on the same datasets. By releasing the GPU implementation code, we expect this work would advance the research on spiking neural networks and neuromorphic computing.

**Acknowledgments**

This material is based upon work supported by the National Science Foundation (NSF) under Grants No.1639995 and No.1948201. This work is also supported by the Semiconductor Research Corporation (SRC) under Task 2692.001. Any opinions, findings, conclusions or recommendations expressed in this material are those of the authors and do not necessarily reflect the views of NSF, SRC, UC Santa Barbara, and their contractors.

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
