[Supplementary Material · NeurIPS_2019_ST_RSBP_Supplementary Material.pdf]

# Supplementary Materials for: Spike-Train Level Backpropagation for Training Deep Recurrent Spiking Neural Networks

**Wenrui Zhang**
University of California, Santa Barbara
Santa Barbara, CA 93106
wenruizhang@ucsb.edu

**Peng Li**
University of California, Santa Barbara
Santa Barbara, CA 93106
lip@ucsb.edu

## 1 Detailed Description of Spike-train Level Post-synaptic Potential (S-PSP) and Total PSP (T-PSP)

S-PSP captures the spike-train level interactions between a pair of pre/post-synaptic neurons and can be defined for any neural models with an all-or-none spiking characteristics and any synaptic models [2]. Without loss of generality, we describe S-PSP using the widely adopted leaky integrate-and-fire (LIF) model of spiking neurons and a first-order synaptic model [1]:

$$\tau_m \frac{u_i(t)}{dt} = -u_i(t) + R\,\alpha_i(t), \tag{1}$$

with

$$\tau_s \frac{\alpha_i(t)}{dt} = -\alpha_i(t) + \sum_j w_{ij} \sum_{t_j^{(f)}} D\left(t - t_j^{(f)}\right), \tag{2}$$

where $u_i(t)$ is the membrane potential of the neuron $i$, $\alpha_i(t)$ the total synaptic current input based on a first order synaptic model with time constant $\tau_s$, and $\tau_m$ the time constant of membrane potential with value $\tau_m = RC$. $R$ and $C$ are the effective leaky resistance and effective membrane capacitance and R is set to 1 since it can be absorbed into synaptic weights. $w_{ij}$ is the weight of the synapse from the pre-synaptic neuron $j$ to the neuron $i$. $t_j^{(f)}$ denotes a particular firing time of the neuron $j$. $D(t)$ is the Dirac delta function.

The integration of (1) and (2) leads to the spike response model (SRM) [1]:

$$u_i(t) = \sum_j w_{ij} \sum_{t_j^{(f)}} \epsilon\left(t - \hat{t}_i^{(f)}, t - t_j^{(f)}\right), \tag{3}$$

where $\hat{t}_i^{(f)}$ denotes the last firing time of the neuron $i$. $\epsilon(s,t)$ specifies the normalized time course of the *post-synaptic potential* evoked by a single firing spike of the pre-synaptic neuron:

$$\epsilon(s,t) = \frac{1}{C} \int_0^s \exp\left(-\frac{t'}{\tau_m}\right) \alpha_i\left(t - t'\right)\, \mathrm{d}t'. \tag{4}$$

Through integration, (4) can be re-written as:

$$\epsilon(s,t) = \frac{e^{(-\max(t-s,0)/\tau_s)}}{1 - \frac{\tau_s}{\tau_m}} \left[e^{\left(-\frac{\min(s,t)}{\tau_m}\right)} - e^{\left(-\frac{\min(s,t)}{\tau_s}\right)}\right] H(s)H(t), \tag{5}$$

where $H(t)$ is the Heaviside step function.

Figure 1: The computation of the S-PSP.

Note that each neuron fires whenever its post-synaptic potential reaches the firing threshold. We now sum up the contributions of the pre-synaptic neuron $j$'s spike train to the (normalized) post-synaptic potential of the neuron $i$ right before all the neuron $i$'s firing times as illustrated in Fig. 1:

$$e_{ij} = \sum_{t_i^{(f)}} \sum_{t_j^{(f)}} \epsilon(t_i^{(f)} - \hat{t}_i^{(f)}, t_i^{(f)} - t_j^{(f)}), \tag{6}$$

defining the (normalized) **spike-train level post-synaptic potential** (S-PSP) from the neuron $j$ to the neuron $i$.

The significance of S-PSPs lies on that it characterizes the aggregated effect of the spike train of the pre-synaptic neuron $j$ on the membrane potential of the post-synaptic neuron $i$ and its firing activities. Employing S-PSPs in the proposed ST-RSBP algorithm is beneficial; it allows efficient consideration of the temporal dynamics and recurrent connections of an RSNN across all firing events at the spike-train level without expensive unfolding in time and backpropagation time point by time point, which are required by BPTT.

The sum of the weighted S-PSPs from all pre-synaptic neurons of the neuron $i$ is defined as the **total post-synaptic potential** (**T-PSP**) $a_i$, relating to the neuron $i$'s firing count $o_i$ via the firing threshold $\nu$:

$$a_i = \sum_j w_{ij} \, e_{ij}, \qquad o_i = g(a_i) \approx \frac{a_i}{\nu}. \tag{7}$$

$a_i$ and $o_i$ are analogous to the pre-activation and activation in the traditional ANNs, respectively, and $g(\cdot)$ can be considered as an activation function converting the T-PSP to the output firing count.

## 2  Detailed Derivation of ST-RSBP Algorithm

The rate-coded loss is defined at the output layer as:

$$E = \frac{1}{2} \|\boldsymbol{o} - \boldsymbol{y}\|_2^2 = \frac{1}{2} \|\frac{\boldsymbol{a}}{\nu} - \boldsymbol{y}\|_2^2, \tag{8}$$

where $\boldsymbol{y}$, $\boldsymbol{o}$ and $\boldsymbol{a}$ are vectors of the desired output neuron firing counts (labels), actual firing counts, and the T-PSPs of the output neurons, respectively. Differentiating (8) with respect to each trainable weight $w_{ij}^k$ incident upon the layer $k$ leads to:

$$\frac{\partial E}{\partial w_{ij}^k} = \frac{\partial E}{\partial a_i^k} \frac{\partial a_i^k}{\partial w_{ij}^k} = \delta_i^k \frac{\partial a_i^k}{\partial w_{ij}^k}, \qquad \text{with} \quad \delta_i^k = \frac{\partial E}{\partial a_i^k}, \tag{9}$$

where $\delta_i^k$ and $\frac{\partial a_i^k}{\partial w_{ij}^k}$ are referred to as the ***back propagated error*** and ***differentiation of activation***, respectively, for the neuron $i$. ST-RSBP updates $w_{ij}^k$ by $\Delta w_{ij}^k = \eta \frac{\partial E}{\partial w_{ij}^k}$, where $\eta$ is a learning rate.

## 2.1 Back Propagated Errors

### 2.1.1 Output Layer

When the layer $k$ is the output layer, the back propagated error at the $i_{th}$ neuron of the layer is given by differentiating the loss defined in (8):

$$\delta_i^k = \frac{\partial E}{\partial a_i^k} = \frac{(o_i^k - y_i^k)}{\nu^k}, \tag{10}$$

where $o_i^k$ is the actual firing count, $y_i^k$ the desired firing count (label), and $a_i^k$ the corresponding T-PSP.

### 2.1.2 Hidden Layers

At each hidden layer $k$, by applying the chain rule, the back propagated error $\delta_i^k$ for the neuron $i$ can be expressed as:

$$\delta_i^k = \frac{\partial E}{\partial a_i^k} = \sum_{l=1}^{N_{k+1}} \frac{\partial E}{\partial a_l^{k+1}} \frac{\partial a_l^{k+1}}{\partial a_i^k} = \sum_{l=1}^{N_{k+1}} \delta_l^{k+1} \frac{\partial a_l^{k+1}}{\partial a_i^k}. \tag{11}$$

$N_{k+1}$ is the number of neurons in the layer $k+1$. Define two error vectors $\boldsymbol{\delta}^{k+1}$ and $\boldsymbol{\delta}^k$ for the two layers: $\boldsymbol{\delta}^{k+1} = [\delta_1^{k+1}, \cdots, \delta_{N_{k+1}}^{k+1}]$, and $\boldsymbol{\delta}^k = [\delta_1^k, \cdots, \delta_{N_k}^k]$, respectively for the layers $k+1$ and $k$, where $N_k$ is the number of the neurons in the layer $k$.

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

$$
\Omega_{ij}^{k+1,k} = \begin{cases} 1 - \frac{1}{\nu^{k+1}} \left( \sum_m^{N_k} w_{lm}^{k+1} \frac{\partial e_{lm}^{k+1}}{\partial o_l^{k+1}} + \sum_p^{N_{k+1}} w_{lp}^{k+1} \frac{\partial e_{lp}^{k+1}}{\partial o_l^{k+1}} \right) & i = j \\ 0 & i \neq j \end{cases}
$$

$$
P_{ij}^{k+1,k} = \frac{\partial a_i^{k+1}}{\partial a_j^k} \qquad \Phi_{ij}^{k+1,k} = w_{ij}^{k+1} \frac{1}{\nu^k} \frac{\partial e_{ij}^{k+1}}{\partial o_j^k} \qquad \Theta_{ij}^{k+1,k} = w_{ij}^{k+1} \frac{1}{\nu^{k+1}} \frac{\partial e_{ij}^{k+1}}{\partial o_j^{k+1}}. \tag{19}
$$

The partial derivatives of the S-PSP with respect to the pre-synaptic and post-synaptic firing counts, i.e. $\frac{\partial e_{ij}^{k+1}}{\partial o_j^k}$ and $\frac{\partial e_{ij}^{k+1}}{\partial o_i^{k+1}}$ as needed in (19) will be determined in Section 2.3. Solving the linear system in (18) gives all $\frac{\partial a_i^{k+1}}{\partial a_j^k}$:

$$
\boldsymbol{P}^{k+1,k} = (\boldsymbol{\Omega}^{k+1,k} - \boldsymbol{\Theta}^{k+1,k})^{-1} \cdot \boldsymbol{\Phi}^{k+1,k}. \tag{20}
$$

Note that since $\boldsymbol{\Omega}$ is a diagonal matrix, the cost in factoring the above linear system can be reduced by approximating the matrix inversion using a first-order Taylor's expansion without performing any matrix factorization.

All $N_k$ errors at the layer $k$ back propagated from the layer $k+1$ per (11) is put into a vector form: $\boldsymbol{\delta^k} = [\delta_1^k, \cdots, \delta_{N_k}^k]$, and is given by:

$$
\boldsymbol{\delta}^k = (\boldsymbol{P}^{k+1,k})^T \cdot \boldsymbol{\delta}^{k+1}, \tag{21}
$$

where $\boldsymbol{\delta}^{k+1}$ is the error vector at the layer $k+1$.

**[Backpropagation from a Hidden Feedforward Layer]** Consider the much simpler case of back-propagating errors from a feedforward layer $k+1$ to its preceding layer $k$. Due to non-existence of recurrent connections in the layer $k+1$, (20) is simplified to:

$$
\boldsymbol{P}^{k+1,k} = (\boldsymbol{\Omega}^{k+1,k})^{-1} \cdot \boldsymbol{\Phi}^{k+1,k}. \tag{22}
$$

Since $\boldsymbol{\Omega}^{k+1,k}$ is diagonal, each $\frac{\partial a_l^{k+1}}{\partial a_i^k}$ can be directly computed:

$$
\frac{\partial a_l^{k+1}}{\partial a_i^k} = \frac{\frac{1}{\nu^k} w_{li}^{k+1} \frac{\partial e_{li}^{k+1}}{\partial o_i^k}}{1 - \frac{1}{\nu^{k+1}} \sum_{p=1}^{N_k} w_{lp}^{k+1} \frac{\partial e_{lp}^{k+1}}{\partial o_l^{k+1}}}. \tag{23}
$$

## 2.2 Differentiation of Activation

Per (9), we derive the differentiation of activation $\frac{\partial a_i^k}{\partial w_{ij}}$ under two cases.

### 2.2.1 Feedforward Layers

For a feedforward layer $k$ and based on (2) of the main paper, differentiation of each activation is given by:

$$
\frac{\partial a_i^k}{\partial w_{ij}^k} = \frac{\partial}{\partial w_{ij}^k} \left( \sum_l^{N_{k-1}} w_{il}^k \, e_{il}^k \right) = e_{ij}^k + \frac{1}{\nu^k} \frac{\partial a_i^k}{\partial w_{ij}^k} \sum_l^{N_{k-1}} w_{il}^k \frac{\partial e_{il}^k}{\partial o_i^k}. \tag{24}
$$

The first term on the right side of (24) reflects the direct dependency of $a_i^k$ on $w_{ij}^k$ while the second term captures the dependency of each S-PSP $e_{il}^k$ on the post-synaptic firing count $o_i^k$, which further depends on $w_{ij}$ according to (12). The derivative $\frac{\partial a_i^k}{\partial w_{ij}^k}$ on the right side of (24) is precisely considered in ST-RSBP. However, HM2-BP [2] does not consider the hidden dependency of $e_{ij}^k$ on $w_{ij}^k$ when deriving (24). As a result, the $\frac{\partial a_i^k}{\partial w_{ij}^k}$ term on the right side of (24) is approximated to $e_{ij}^k$.

(24) gives the desired differentiation of activation as:

$$\frac{\partial a_i^k}{\partial w_{ij}^k} = \frac{e_{ij}^k}{1 - \frac{1}{\nu^k}\sum_l^{N_{k-1}} w_{il}^k \frac{\partial e_{il}^k}{\partial o_i^k}}.\tag{25}$$

### 2.2.2 Recurrent Layers

For the activation $a_i^k$ of the neuron $i$ at the recurrent layer $k$, we further consider the recurrent connections and get

$$\frac{\partial a_i^k}{\partial w_{ij}^k} = \frac{\partial}{\partial w_{ij}^k}\left(\sum_l^{N_{k-1}} w_{il}^k\, e_{il}^k + \sum_p^{N_k} w_{ip}^k\, e_{ip}^k\right) = e_{ij}^k + \frac{\partial a_i^k}{\partial w_{ij}^k}\frac{1}{\nu^k}\left(\sum_l^{N_{k-1}} w_{il}^k \frac{\partial e_{il}^k}{\partial o_i^k} + \sum_p^{N_k} w_{ip}^k \frac{\partial e_{ip}^k}{\partial o_i^k}\right),$$

leading to:

$$\frac{\partial a_i^k}{\partial w_{ij}} = \frac{e_{ij}^k}{1 - \frac{1}{\nu^k}\left(\sum_l^{N_{k-1}} w_{il}^k \frac{\partial e_{il}^k}{\partial o_i^k} + \sum_p^{N_k} w_{ip}^k \frac{\partial e_{ip}^k}{\partial o_i^k}\right)}.\tag{26}$$

### 2.3 Differentiation of S-PSP w.r.t Pre/Post-Synaptic Firing Counts

Before presenting the final ST-RSBP algorithm, we shall determine the partial derivatives $\frac{\partial e_{ij}}{\partial o_j}$ and $\frac{\partial e_{ij}}{\partial o_i}$ of an S-PSP $e_{ij}$ with respect to the firing counts of the pre-synaptic neuron $j$ and post-synaptic neuron $i$, respectively, as needed in (19), (23), (25), and (26). As discussed in Section 1, S-PSPs serve as a bridge between neuron-level firing timings and spike-train level firing count and allow backpropagating errors defined for a rate-coded loss at the spike-train level.

In [2], the HM2-BP computes the two partial derivatives by assuming that each S-PSP $e_{ij}$ is approximately linear in both $o_j$ and $o_i$. To examine this assumption, we evaluate the S-PSP from the neuron $j$ to neuron $i$ via a synapse. The LIF neuron model of (1) and the synaptic model of (2) with $\tau_m = 64ms$, $\tau_s = 8ms$ are adopted in this analysis. The simulation duration is set to $600ms$ and the first-order Euler method with a fixed stepsize of $1ms$ is used for simulation. To cover a wide range of interactions between the two neurons, we consider all combinations of the firing rates of two neurons $o_i$ and $o_j$ when they are swept widely from 1 to 50. For each combination of $o_i$ and $o_j$ values, we generate the spike trains of the two neurons by randomly choosing $o_i$ and $o_j$ numbers of random spiking times, respectively, and compute the S-PSP $e_{ij}$ according to (6). We repeat this process 500 times and take the average value of $e_{ij}$.

We plot the relation between the pre/post-synaptic firing counts $o_j$ and $o_i$ and the average $e_{ij}$ in Fig. 3A. Fig. 3B shows that with $o_i$ fixed $e_{ij}$ increases rather linearly in $o_j$, consistent with [2], and hence we have:

$$\frac{\partial e_{ij}}{\partial o_j} \approx \frac{e_{ij}}{o_j}.\tag{27}$$

However, Fig. 3C shows that with $o_j$ fixed, $e_{ij}$ is not linear in a wide range of $o_i$, suggesting that the assumption made in [2] can lead to errors when the postsynaptic firing rates vary a lot. Based on the data collected for Fig. 3A, for each fixed $o_j$, we instead fit $e_{ij}$ as a third-order polynomial in $o_i$ to obtain the corresponding values for the derivative $\frac{\partial e_{ij}}{\partial o_i}$. The characterization of $\frac{\partial e_{ij}}{\partial o_i}$ occurs offline prior to the training process. In this approach, ST-RSBP can more precisely measure the differentiation of S-PSP w.r.t firing counts than HM2-BP [2]. Therefore, ST-RSBP may achieve better results even on feedforward networks like spiking CNNs.

### 2.4 The Final Proposed ST-RSBP Algorithm

For each layer $k$, denote the error vector by $\boldsymbol{\delta}^k \in \mathbb{R}^{N_k}$, the matrix of differentiation of activation by $\boldsymbol{F}^{k,k-1} \in \mathbb{R}^{N_k \times N_{k-1}}$, and the weight matrix from the layer $k-1$ to layer $k$ by $\boldsymbol{W}^{k,k-1} \in \mathbb{R}^{N_k \times N_{k-1}}$, respectively. $\boldsymbol{P}^{k+1,k} \in \mathbb{R}^{N_{k+1} \times N_k}$ contains all derivatives of $\frac{\partial a_l^{k+1}}{\partial a_i^k}$ obtained from (20) or (23). If the layer $k$ is recurrent layer, we additionally use $\boldsymbol{F}^{k,k} \in \mathbb{R}^{N_k \times N_k}$ and $\boldsymbol{W}^{k,k} \in \mathbb{R}^{N_k \times N_k}$ to denote the matrix of differentiation of activation and the weight matrix of recurrent connections within the

Figure 3: (A) The average S-PSP value vs. pre and post-synaptic firing counts; (B) The average $e_{ij}$ vs. $o_j$ when the post-synaptic firing count is fixed ($o_i = 10$); (C) The average $e_{ij}$ vs. $o_i$ when the pre-synaptic firing count is fixed ($o_j = 10$).

layer $k$. Putting everything together, the complete ST-RSBP algorithm with a learning rate $\eta$ is as follows:

$$
\begin{cases}
\Delta \boldsymbol{W}^{k,k-1} = \eta \frac{\nabla E}{\nabla \boldsymbol{W}^{k,k-1}} = \eta \cdot diag(\boldsymbol{\delta}^k) \cdot \boldsymbol{F}^{k,k-1} & \text{for feedforward connections} \\
\Delta \boldsymbol{W}^{k,k} = \eta \frac{\nabla E}{\nabla \boldsymbol{W}^{k,k}} = \eta \cdot diag(\boldsymbol{\delta}^k) \cdot \boldsymbol{F}^{k,k} & \text{for recurrent connections}
\end{cases}
$$

$$
F_{ij}^{k,k-1} = \frac{e_{ij}^k}{1 - \frac{1}{\nu^k} \sum_l^{N_{k-1}} w_{il}^k \frac{\partial e_{il}^k}{\partial o_i^k}}
$$

$$
F_{ij}^{k,k} = \frac{e_{ij}^k}{1 - \frac{1}{\nu^k} \left( \sum_l^{N_{k-1}} w_{il}^k \frac{\partial e_{il}^k}{\partial o_i^k} + \sum_p^{N_k} w_{ip}^k \frac{\partial e_{ip}^k}{\partial o_i^k} \right)}
$$

$$
\begin{cases}
\delta_i^k = \frac{o_i^k - y_i^k}{\nu^k} & \text{if layer k is the output} \\
\boldsymbol{\delta}^k = (\boldsymbol{P}^{k+1,k})^T \cdot \boldsymbol{\delta}^{k+1} & \text{if layer k+1 is feedforward} \\
\boldsymbol{\delta^k} = ((\boldsymbol{\Omega}^{k+1,k} - \boldsymbol{\Theta}^{k+1,k})^{-1} \cdot \boldsymbol{\Phi}^{k+1,k})^T \cdot \boldsymbol{\delta^{k+1}} & \text{if layer k+1 is recurrent}
\end{cases} \tag{28}
$$

$$
\Omega_{ij}^{k+1,k} = \begin{cases}
1 - \frac{1}{\nu^{k+1}} \left( \sum_m^{N_k} w_{lm}^{k+1} \frac{\partial e_{lm}^{k+1}}{\partial o_l^{k+1}} + \sum_p^{N_{k+1}} w_{lp}^{k+1} \frac{\partial e_{lp}^{k+1}}{\partial o_l^{k+1}} \right) & i = j \\
0 & i \neq j
\end{cases}
$$

$$
\Phi_{ij}^{k+1,k} = w_{ij}^{k+1} \frac{1}{\nu^k} \frac{\partial e_{ij}^{k+1}}{\partial o_j^k} \qquad \Theta_{ij}^{k+1,k} = w_{ij}^{k+1} \frac{1}{\nu^{k+1}} \frac{\partial e_{ij}^{k+1}}{\partial o_j^{k+1}}
$$

$$
P_{ij}^{k+1,k} = \frac{\frac{1}{\nu^k} w_{ij}^{k+1} \frac{\partial e_{ij}^{k+1}}{\partial o_j^k}}{1 - \frac{1}{\nu^{k+1}} \sum_{p=1}^{N_k} w_{ip}^{k+1} \frac{\partial e_{ip}^{k+1}}{\partial o_i^{k+1}}}.
$$

The application of ST-RSBP follows the typical backpropagation steps. First, the SNN is simulated layer-by-layer based on chosen synaptic/neural models such as the LIF model (1). Second, the firing counts of the output layer are compared with the desirable firing labels to compute the output error $\boldsymbol{\delta^k}$. After that, the error vector in the output layer is propagated backwards to determine the gradient, based on which both the recurrent synapses weights and the weights between layers are trained.