[Reviews · NeurIPS 2019]

Reviewer 1



Mein Concerns: The main motivation of the paper, to solve Backprop in spiking neurons, is not an open problem in computational neuroscience. In fact, learning in spiking neural networks using standard methods is not a problem at all as recent work shows. It has been demonstrated multiple times that Backprop can be applied without much changes by applying pseudo-derivatives to circumvent the non-differentiable spikes. See: [6-8]. This works very well in practice and scales up to midscale benchmark problems (and possibly beyond) without performance loss compared to classical (analog) neural networks. In this context it hard to pinpoint the main innovation of the manuscript. The presentation of the model is mixing existing ideas with details that are original to the present paper. For example the outline of the spiking backpropagation in Figure 2 is very hard to decode. It is unclear what the dashed lines represent without additional knowledge. A caption, labels and/or legend would help. The figure is also not sufficiently explained in the main text. The derivations on page 4-6 could be compressed a great deal since they are very standard. Additional details could be pushed to a supplement making space for the relevant details. Finally it is unclear why the proposed model is more biologically plausible than previous models. The learning rules that are presented are applications of standard Backprop to spiking networks. This was shown before. For example, also the model presented in ref. 4 in the manuscript could be applied to spiking recurrent networks (not just LSTMs). Other recent approaches that use feedback weights (synthetic gradients) for deep learning in SNNs [1,2] and their recent extension to recurrent neural networks [3], don't need unrolling the network over time. These prior related work should be discussed and cited. What is the advantage of the proposed model compared to this prior related work? References: [1] Lillicrap, T. P., Cownden, D., Tweed, D. B., and Akerman, C. J. (2016). Random synapticfeedback weights support error backpropagation for deep learning.Nature communications,7:13276. [2] Samadi, A., Lillicrap, T. P., and Tweed, D. B. (2017). Deep learning with dynamic spikingneurons and fixed feedback weights.Neural computation, 29(3):578–602. [3] Biologically inspired alternatives to backpropagation through time for learning in recurrent neural nets Guillaume Bellec, Franz Scherr, Elias Hajek, Darjan Salaj, Robert Legenstein, Wolfgang Maass. https://arxiv.org/abs/1901.09049

Reviewer 2



The paper introduces a method of applying back propagation for fitting parameters of spiking neural networks. The approach relies on firing rates of spike trains to define the errors and uses what seems fairly similar to standard backpropagation with some exception for the recurrent case. I find two main issues in the paper: 1. In event based processing, you typically want to treat events dynamically. The proposed algorithm treats a chunk of data at a time since it needs a history of spikes to define the error and therefore is not really event based. 2. This point is probably rephrasing the former in some way. Essentially it appears that the algorithm is not a new back propagation algorithm but rather a new model for spike data that allows for processing with standard back-propagation (except maybe in the recurrent case). Therefore, in terms of separating data model from optimization/fitting method the work is not presented clearly. Questions: You cite the paper [23, 38, 4, 33] for providing competitive performance using a similar approach to yours however it appears that you do not compare with any of them. Why is that? Please clarify or modify. Minor Comments: Probably the exposition of your algorithm can be simplified by some restructuring so as to avoid superscripts/subscripts. It just seems too verbose at the moment. Update after review: The rebuttal clarifies considerably both in terms of a comparison with BPTT and for the event based processing. I will modify the score accordingly

Reviewer 3



The paper addresses the problem of training spiking neural networks (SNNs), in particular recurrent SNNs, with a new form of backpropagation (BP). BP on SNNs is hard because of temporal effects and the discontinuous nature of spikes. The topic of deriving BP rules for SNNs has received increasing attention in the previous years, and the most relevant works are referenced. The present paper is valuable in extending the scope to recurrent networks, and giving more attention to temporal effects, which are neglected in other approaches. So the topic itself is not original, but it is a valuable extension of the state-of-the-art. The main contribution is the ST-RSBP learning rule, which backpropagates over spike trains, rather than unrolling recurrent networks completely over time. Spike-train level Post-synaptic potentials (S-PSPs) are introduced, which accumulate the contributions of the pre-synaptic neuron to the PSP right before the spike-time of the post-synaptic neuron. From this, an approximation of the spike count is derived, which in turned is used as the basis for the backpropagation derivation. The following remains unclear to me: - The BP algorithm requires a desired output firing count y (in (3)). How is this determined? Is there a constant firing rate for the correct output assumed, and if yes, what is it? - When is the BP update performed? Is it a batch-mode update, or can online learning be performed as well? - How suitable is the learning rule for hardware implementations of SNNs in neuromorphic or digital platforms? - How are firing thresholds set? Section 4.1 says it is "depending on the layer", but does not give more results. Experiments are performed on 4 datasets, where in every case an improvement over previous BP-based SNN training methods is shown. There is also a comparison to non-spiking approaches, but I am not sure how the authors picked the references. For example for Fashion MNIST there are much better solutions available than the ones reported here, there are even multiple tutorials that show how to reach 93% or more. - I strongly recommend also reporting the non-spiking state of the art for each example to give a fair comparison and not over-sell the SNN results. - I am not sure why a recurrent model was chosen for the (static) Fashion MNIST dataset. Overall I think this is a good contribution, and the presentation is OK, although a lot of the derivations can only be understood with the help of the supplementary material. Recurrent SNN training is not a major topic at NeurIPS, but is of interest to the community. As a final suggestion I would recommend re-working the abstract, because it tells a lot of very generic facts before discussing the actual contributions of the paper. =================== Update after author feedback: I want to thank the authors for addressing my and the other reviewers' questions in their author feedback. My main concerns have been addressed, and I think this paper should be accepted, accordingly I will raise my score to 8. Still, I think there are some improvements to be made, as addressed by the reviewer concerns and the author feedback: - please make explicitly clear that you are not addressing CNN-type networks - I think it would be great if you can include an outlook on the hardware implementation on FPGAs, because I think this is an important point for this rule - Please make the original contributions compared to existing spiking backprop variants clearer - Please include all hyperparameters (thresholds, learning rates, ...) also in the supplementary material, and not just in the code.

[Author Response · NeurIPS 2019]

We deeply appreciate and will address all insightful review comments in final paper with major ones responded below:

**1. Main contributions:** ST-RSBP can transparently train all types of SNNs including RSNNs without unrolling in time. The employed S-PSP model improves training efficiency at the spike-train level and also addresses discontinuity of spiking activity for accurate gradient computation. The spike-train level processing for RSNNs is the starting point for ST-RSBP. After that, we have applied the standard BP principle while dealing with specific issues of derivative computation at the spike-train level. Unlike methods such as Feedback Alignment, Direct Feedback Alignment, and e-prop, ST-RSBP is not biologically plausible - a limitation. Biologically plausible methods tend to produce somewhat lower performance; ST-RSBP trades off biological plausibility for performance.

**2. Comparison with BPTT:** We revise Figure 2 on the right to better illustrate the difference between the proposed ST-RSBP and other BPTT based rules. BPTT first unfolds a RSNN in time to effectively remove recurrent connections and then backpropagates the error across the whole unfolded network and along the discretized time points, during which non-differentiability of spiking activity must be dealt with. ST-RSBP operates on the spike-train level, preforms training while avoiding unfolding the RSNN and backpropagating through the long unfolded path.

**3. Comparison with other works:** Table 1 compares the proposed ST-RSBP with other works on N-MNIST, the well known neuromorphic version of MNIST. Since none of these works has been tested on RSNNs, we only compare the results on feed-forward SNNs. As shown in Table 1, ST-RSBP outperforms the BPTT based rules [23, 38] and is slightly better than [19, 33]. Moreover, ST-RSBP is readily applicable to RSNNs. In Table 2, we evaluate ST-RSBT using the Sequential MNIST dataset under the RSNN setting based on the same preprocessing of [4]. The LIF network in [4] is a fully-connected RSNN without the special adaptive neurons proposed in [4] and is trained using BPTT. We test ST-RSBP on fully-connected RSNNs with a size equal to or smaller than that of the LIF network. Table 2 shows that the proposed ST-RSBP outperforms the BPTT adopted in [4]. Note that the main contribution of [4] is on a new type of SNNs, namely Long Short-Term Memory Spiking Neural Networks (LSNNs) with the special adapting neurons, demonstrating very good performance. Our comparison here only intends to show that from a training perspective, ST-RSBP outperforms BPTT when training similar standard RSNNs. We expect that by modifying our ST-RSBP rule we can also train LSNNs to enhance training quality.

Table 1: Performance on N-MNIST

| Model | Hidden layers | Accuracy |
|---|---|---|
| Spiking MLP[23] | 800 | 98.74% |
| STBP[38] | 800 | 98.78% |
| HM2BP[19] | 800 | 98.88% |
| SLAYER[33] | 500-500 | 98.89% |
| ST-RSBP | 800 | **98.91%** |

Table 2: Performance on Sequential MNIST

| Model | Hidden layers | Accuracy |
|---|---|---|
| LIF[4] | R220 | 63.30% |
| ST-RSBP | R128 | **76.52%** |
| ST-RSBP | R220 | **77.39%** |

**4. Event-based Processing, Hardware-Friendliness, Implementation Settings, and Fashion MNIST**

ST-RSBP performs supervised training and only updates the weights at the end of the spike train of each example when the loss is available. However, the computation of S-PSPs, the main overhead fo ST-RSBP, can be accumulated spike-by-spike in an event-driven online manner in the forward pass of BP, removing the need of storing the spiking history of the network. This feature makes ST-RSBP amenable to neuromorphic hardware implementation. Recently, we have successfully demonstrated online S-PSP computation on FPGA for a different training algorithm.

The parameters like the desired output firing counts, thresholds, learning rates are empirically tuned. The chosen values for each network reported in Section 4 are summarized in the "Results" directory of the source code repository.

Non-spiking ANNs that produce better results than ST-RSBP on Fashion-MNIST are different types of CNNs. We do not evaluate ST-RSBP on spiking CNNs. In Table 4, we only compare ST-RSBP with the best performing methods on non-CNN feedforward networks including BP for ANNs. Training spiking CNNs using BP is very time consuming. In the future, we will demonstrate the application of ST-RSBP to spiking CNNs.

For demonstration purpose, we adopt the recurrent model for Fashion MNIST to show the ability of ST-RSBP to train RSNNs on different datasets. Here we also use ST-RSBP to train a 400-400 feed-forward SNN with accuracy of 90.08% on Fashion MNIST, also surpassing all other methods in Table 4 of the paper.

[Meta-Review · NeurIPS 2019]

The authors propose a variant of the backpropagation through time (BPTT) algorithm for spiking neural networks (SNNs). An interesting aspect is that, instead of unrolling the network computation over time, backpropagation over spike trains is performed. The algorithm is tested on various datasets, achieving state-of-the-art results for SNNs. The approach is very original and innovative. The results are very good and of interest for the community interested in spiking neural networks. We therefore agreed that the manuscript is suitable for publication at NeurIPS. The authors are advised to re-work the abstract and introduction. There, one gets the impression that the algorithm is biologically plausible, which is not the case according to the Author's feedback.